# Influence of Reaction Parameters on the Gelation of Silanised Linseed Oil

**DOI:** 10.3390/ma13235376

**Published:** 2020-11-26

**Authors:** Ewelina Depczyńska, Waldemar Perdoch, Bartłomiej Mazela

**Affiliations:** Faculty of Forestry and Wood Technology, Poznań University of Life Sciences, 60-637 Poznan, Poland; waldemar.perdoch@up.poznan.pl (W.P.); bartlomiej.mazela@up.poznan.pl (B.M.)

**Keywords:** bio-based materials, linseed oil, silane, silylation reaction, VTMOS

## Abstract

The subject of this work was to characterize the catalytic course of the linseed oil silylation reaction with vinyltrimethoxysilane (VTMOS), carried out under elevated pressure and temperature conditions, and an explanation of the reasons for rapid gelation of the reaction product. To explain and describe the process, analytical methods were used, i.e., ^1^H and ^13^C NMR (nuclear magnetic resonance), GC-FID (gas chromatography coupled with flame ionisation detection), and GPC (gel permeation chromatography). Reaction products were monitored after 3, 6 and 12 h. The molar mass of the VTMOS-modified oil in only 3 h was comparable with the molar mass of the product obtained by conventional polymerisation. An increase in the reaction time resulted in further transformations resulting from the hydrolysis and condensation reactions taking place. In contrast to reactivity of soybean oil, the silanisation of linseed oil occurred much faster and without the need for cross-linking catalysts. The reason for the high reactivity of linseed oil to VTMOS and rapid gelation of the resulting product was primarily the amount of double bonds present in linseed oil and their high availability, in particular the double bond in the acid linolenic acid located at the C16 carbon.

## 1. Introduction

The literature on the subject explicitly indicates that in the 21st century, triglyceride oils can be treated as the most important raw materials for the synthesis of polymers produced from renewable resources [1]. The polymers are prepared by means of various strategies, the choice of which determines the direction of polymerisation. The presence of the oil/fatty acid chain in the polymer structure improves some of the polymer physical properties in terms of their flexibility, adhesion, resistance to water and chemicals resistance. Due to their source and structural nature, triglyceride oils can be also widely used as pure polymers. Factors such as biocompatibility and/or biodegradability make triglyceride oils indispensable as raw materials in modern industrial solutions. Currently, the largest area of triglyceride oils’ application is concentrated in the field of coatings, including starting raw materials for the synthesis of other polymers, intended for varnish products [2]. The most commonly used oils in the range of application are: soybean [3], linseed and tall [4,5], coconut [6], oiticica [7], cardanol [8], tung [9], yellow oleander seeds oil [10], from Ricinodendron heudelotii, palm [11,12], castor oil [13], and from Nahar seeds [14]. To enhance the range and effectiveness of the oils’ application, it seems necessary to polymerize and modify them. Radical polymerisation is one of the common methods used to modify oils [1]. The course of this reaction is shown in Figure 1. Although the radical polymerisation of linseed oil is a process known and widely described in the literature, its kinetics is not fully understood, owing to the complexity of the sequential stages. The radicals arising from the influence of the thermal energy or UV/VIS (Ultraviolet/Visible Spectroscopy) and an explanation of the reasons for rapid gelation of the reaction product radiation are responsible for initiating the reaction. As a result of the energy supplied, vibrations in the molecule are increased and the multiple bonds break up with the formation of radicals [15]. Copolymerisation is one of the common methods used to modify triglyceride oils. The first oxidation step involves separating hydrogen from the methylene group between double bonds in the polyunsaturated fatty acid chain [16], and radical recombination leads to crosslinking of structures and the formation of alkyl, ether or peroxide bridges [17,18,19]. Peroxide bridges decompose to alkoxy radicals [20]. Oxidized oils are broadly employed in the production of binders based on natural oils, as they give end products with high viscosity and good film formation properties.

Various attempts to modify natural oils in order to use them in wood preservatives are reported in the literature on the subject. Veigel et al. [21] recommended a solution based on the modification of linseed oil with nano-cellulose fibres. In this way, an obtained product increased its mechanical parameters (resistance to abrasion) and resistance to weather conditions after application on wood. Other solutions consisted of urethanising [22,23] or epoxidation of oil in situ, followed by copolymerisation with vinyl acetate [24]. Wood impregnated with epoxied oil after copolymerisation showed enhanced resistance to some types of wood decaying fungi. Srinivasan [25] and Zhuang [26] analysed the reactions of soybean oil with silanes (e.g., VTMOS—vinyltrimethoxysilane) and their possible courses. As far as vinyltrimethoxysilane (VTMOS) reactions with oils are concerned, the authors implied they are most likely to proceed according to Alder’s “ene” reaction. The reaction occurs under elevated temperature (above 200 °C), at elevated pressure in the presence of a 2.5-bis (tert-butylperoxy)-2.5-dimethylhexane catalyst. The reaction occurs spontaneously in the gas phase. However, the higher the temperature and the longer the reaction time, the higher the efficiency of the silane group substitution to the unsaturated fatty acid. Tambe et al. [27] conducted similar reactions not only for soybean oil, but also for rapeseed and abyssinic oils. The soybean oil and VTMOS reaction mechanism is shown in Figure 2.

Han et al. [28] and Schneider [29] also described the silanisation reaction mechanism and a double bond reactivity in silanes (e.g., VTMOS). In addition to the course of the reaction, the reason for the hydrolysis of the polymer was also studied, and was explained by a small amount of added photoinitiator or a long reaction time [30]. After introducing water and cure coating accelerator (dibutyltin dilaurate—DBTDL) to the reaction products shown in Figure 2, hydrolysis was observed followed by condensation leading to crosslinking and curing of the coating, as well as obtaining more complex polymer structures later (Figure 3). This process was not observed in the reaction conducted without a catalyst.

Depczyńska et al. [31], on the basis of monitoring the pressure during the linseed oil reaction, VTMOS and the 2.5-bis (tert-butylperoxy)-2.5-dimethylhexane catalyst, spectral and chromatographic analysis, found that new polymers were obtained as a result of the synthesis. Additionally, it was found that the viscosity of the obtained reaction products was significantly lower than the viscosity of unmodified polymerised linseed oil. However, the authors did not explain the reasons for gelling the silanised product of modified linseed oil.

The subject of this work was to characterise the catalytic course of the linseed oil silylation reaction with vinyltrimethoxysilane, carried out under elevated pressure and temperature conditions. The authors attempted to explain the cause of rapid gelation of the silanised linseed oil.

## 2. Materials and Methods

### 2.1. Materials

Linseed oil was purchased from Alberdingk boley gmbh (Krefeld, Germany). The vinyltrimethoxy silane (VTMOS) [CAS 2768-02-7] with purity 99% was purchased from Dow Corning (Midland, MI, USA). The catalyst 2.5-Bis(tert-butylperoxy)-2.5-dimethylhexane, (Luperox 101 [CAS 78-63-7]) was purchased from Arkema (Colombes, France).

### 2.2. Silylation Reaction

The reaction of oil, VTMOS and catalyst was conducted in a high-pressure reactor (PARR Instrument Company, Moline, IL, USA). Prior to silylation, the heating reactor was purged with nitrogen. The reaction components were put into the reactor and after 5 min of the mixture stirring at room temperature, heating was started. Depending on the substrates, proper reaction temperature was achieved after 90–120 min. The reaction was run at a temperature of 280 °C with a screw speed of 220 rpm ± 2 rpm throughout the entire process. The composition of the reaction mixture and the amount of the components inserted into the reactor are summarized in Table 1.

### 2.3. Characterisation of Silylated Oil

The reaction products were described by the following technical parameters: acid number; gel permeation chromatography (GPC); gas chromatography with flame ionisation detector (GC-FID); proton and carbon nuclear magnetic resonance (^1^H NMR, ^13^C NMR).

#### 2.3.1. Acidic Number of Silylated Oil

The acid number was determined according to PN-EN ISO 2114:2005 Plastics (polyester resins) and paints and varnishes (binders)—Determination of partial acid value and total acid value (ISO 2114:2000). The acid number was determined according to the following formula:(1)$$LK=\frac{\left[56.1\;\left(V2\;-\;V1\right)c\right]}{m}$$
where:
LK—acid number (mg KOH/g),56.1—constant value (molar mass KOH) (g/mol),c—concentration of potassium hydroxide solution (mol/l),V1—volume of titrant consumed in the blank test (ml),V2—volume of titrant consumed in the determination (ml),m—mass of the analytical sample.

Each silanised product was subjected to the following measurements: GC-FID analysis, polydispersity; NMR (^1^H, ^13^C) analysis. The tests allowed the estimation of the change in free fatty acids content in a final product.

#### 2.3.2. GC-FID Analysis

The silanol products were analysed by GC-FID in order to quantify the amount of reacted VTMOS. The analysis was carried out with the use of GC7890 Agilent Technologies (Santa Clara, CA, USA) gas chromatograph apparatus with a flame ionisation detector.

#### 2.3.3. Polydispersity Index

The polydispersity index (PDI) and the average molecular weight number of the modified oil were estimated through the GPC analysis. GPC analyses with high-performance liquid chromatography (HPLC) with refractive index detection Waters 2690/95 (Milford, MA, USA) liquid chromatograph. Polymers were soluble in THF (tetrahydrofuran) with GPC. Results were calculated with polystyrene standards. Toluene was used as a retention time marker. The assay was made to determine the molar mass of new polymers and the degree of uniformity in the post-reaction mixture composition. According to the literature, PDI for polymers takes the values shown in Table 2:(2)$$\frac{Mw}{Mn}>1$$
where:
Mw > MnMn—average molar mass (by number) (g/mol)Mw—average molar mass (mass) (g/mol)

**Table 2 materials-13-05376-t002:** PDI value for polymers.

PDI	Polymer Description
1.0	Początek formularza hypothetical monodisperse polymer, living polymers (almost monodispersive) Dół formularza
1.5–2.0	addition polymers
<5.0	polymers with a low molecular weight distribution
5.0–20.0	polymers with an average molecular weight distribution
>20	polymers with a high molecular weight distribution
8.0–30.0	coordination polymers
20.0–50.0	branched polymers

#### 2.3.4. Nuclear Magnetic Resonance (^1^H, ^13^C)

The structure of the products was determined with ^1^H and ^13^C NMR using an NMR Bruker Avance III 600 MHz (Billerica, MA, USA). All spectra were made in deuterated chloroform (CDCl_3_) at a sample concentration of several dozen mg/mL (concentration would not be precisely determined). For ^1^H NMR spectra, the resonant frequency for the hydrogen nucleus characterising the spectrometer on which spectra were made is 400 MHz, the number of performed scans was 32. For the ^13^C NMR spectra, the above values were 150 MHz and 1024 scans, respectively. In addition, ^13^C NMR spectra were made using spin decoupling on channel ^1^H.

## 3. Results

### 3.1. The Acid Number Determination

The results of determining the acid number of oil and oil after modification are summarized in Table 3. It was observed that oil polymerisation, in particular in the presence of Luperox 101 (Colombes, France) catalyst (Reaction 2), significantly increased the determined value, the number of carboxyl groups, which in the tested product was over 10 times higher than in raw oil. As a result of the VTMOS oil modification, the number of carboxylic groups in the product, relative to the number of carboxylic groups determined for unmodified oils, has not altered. These observations confirmed that a new product was created as a result of the silanisation, which in terms of basic technical parameters differed from the product resulting from the commonly used polymerisation.

### 3.2. GC-FID Analysis

The content of VTMOS determined in the selected reaction stages is summarized in Table 4. It is obvious that the marked VTMOS content in reactions 1 and 2 (< 0.02%) resulted from imperfections of the method and was treated as a silane content originating from the so-called measurement background. After the first three hours of oil modification, only 5% of VTMOS, relative to the initial silane amount, was identified in the tested product. This observation indicates the reaction or degradation of the silane in the reaction mixture. Silanisation of the oil during 6 h (reaction 3) led to gelation of the product. It seems significant that linseed oil modified without the presence of a catalyst also gelled after 6 h. The structure was crosslinked despite the lack of crosslinking agents described in the literature, i.e., water and dibutyltin dilaurate.

### 3.3. GC-FID Analysis

The average molar mass (Mw) of modified linseed oil following 3 h of reaction at 280 °C was about 40% lower than for unmodified polymerised oil after 12 h. After more than 3 h, the product gelled and the molar mass exceeded 20,000 g/mol. The polymer from the reaction 3 was characterised by the highest molecular weight distribution. The polydispersity index was much higher than in products from other reactions. Polydispersity at the level 15.52 allowed classifying the resulting polymer to the medium range of a molar mass distribution (Rąbek 2008). In the reaction carried out on the partially polymerised linseed oil (reaction 4), a molar mass similar to the unmodified 12 h polymerisation oil was obtained (reaction 1). However, the polydispersity index was higher, which indicates a different chain length of the resulting macromolecule.

Moreover, the catalyst effect on the reaction of oil with VTMOS was determined. The absence of a catalyst and the high temperature reaction led the product to gel. The molar masses were low, indicating the breakdown of molecules. The above observations suggest that VTMOS block the standard radical polymerisation reaction of linseed oil. The average molar mass of the product was about 50% lower than of raw linseed oil. Based on the described observation, it can be inferred that pure VTMOS caused the chains’ breakdown of compounds contained in linseed oil, and no polymerisation occurred during the thermal treatment process. The detailed results of the GPC analysis are summarized in Table 5.

### 3.4. NMR Spectrum Analysis

On the basis of a spectrum and literature analysis [25,27], separate peaks originating from linseed oil (Figure 4 and Figure 5), VTMOS (Figure 6) and silanisation products (Figure 7, Figure 8, Figure 9 and Figure 10) were determined. In the ^1^H NMR spectra of polymerised linseed oil from synthesis 4, a reduction in the intensity of peaks from groups with a double bond was observed, which confirmed the radical polymerisation process (Figure 7). In the ^1^H NMR spectrum of the silane, peaks were present at 3.55 ppm and 5.96 ppm, which stemmed from the protons of the methoxy group (-OCH_3_) and from the protons of the vinyl group (-CH=CH_2_), respectively. In the ^1^H NMR spectrum of the modified linseed oil samples, a peak at 3.65 ppm was present which derived from the methoxy (-OCH_3_) silane group protons. On the spectrum, the peak at 5.98 ppm from the vinyl group protons (-CH=CH_2_) was not observed, which indicates that the silane reacted with the oil. In the spectrum of the gelled product (Figure 8), the intensity of the peaks from the double bond groups is reduced, which confirms the oil polymerisation reaction and the reaction between the silane and the oil. In the ^1^H NMR spectrum an increased intensity of methoxy groups in the 3.66 ppm range is observed, which confirms the hydrolysis and condensation of the polymer leading to the observed effect of gel formation.

In the ^13^C NMR spectrum of modified linseed oil (Figure 9), a peak around 50 ppm originating from the methoxy group (-OCH_3_) was observed. In the same spectrum, no peak was observed at around 138 ppm due to the vinyl group (-CH=CH_2_). These observations indicate the reaction of the silane with oil and the lack of free silane in the reaction product. As in the spectrum ^1^H NMR, a decrease in the intensity of peaks stemming from the double bond (C=C) was observed, which confirms the modification of the oil, i.e., the reaction between the silane and the oil. In the ^13^C NMR spectrum of reaction 3 product (gelled oil), the increased intensity of methoxy groups in the 51.37 ppm range and the acyl groups in the area 14.10–34.00 ppm indicates the hydrolysis and condensation of the polymer (Figure 10).

## 4. Discussion

The difference in the reactivity of linseed oil and that of soybean oil widely described in the literature [27] can be seen in their composition, and particularly in the content of fatty acids. Soybean oil contains only 7% Linolenic acid whereas linseed oil does not exceed 52% [31]. Linolenic acid contains up to three double bonds, one of which, located at C_16_ carbon, is more exposed than others. Less steric hindrance makes the bonds more susceptible to radical polymerisation. In Linoleic acid, which is 53% soybean oil, the most reactive double bond is at the C_13_ carbon atom attached to the n-pentane group (steric hindrance for radical polymerisation). It is closely related to the range of free radical activities. Since C_16_ carbon in Linolenic acid of linseed oil is involved in the polymerisation reaction, the VTMOS-fatty acid reaction can take place on the unconjugated C_9_ carbon atom or on the C_16_ atom by blocking the polymerisation. GPC analysis confirmed the formation of polymers; therefore, it is very likely that VTMOS attacks the double bond with C_9_ Linolenic acid according to the “Aldera-ene” mechanism. VTMOS can also react with Linoleic acid (about 17%) in linseed oil (similar to soybean oil [27]). However, if this was the only VTMOS reaction with the fatty acid chain, the linseed oil polymerisation should occur in a manner similar to traditional polymerisation, and consequently the reaction product would also be similar.

Additionally, at elevated temperatures, the polymerisation proceeds faster, and its termination may occur via recombination of the macro-radical. Macro-gel is formed because the microgels formed at the polymerisation initial stage polymerize with each other by means of unreacted double bonds. The high temperature affects the degree of polymer branching. In the studied case, the methoxysilane groups may react with the remaining double bonds in the fatty acid chain. Based on the conducted analyses and observations, a catalytic polymerisation reaction mechanism conducted under conditions of the elevated pressure and temperature was recommended (Figure 2).

## 5. Conclusions

The results of the GPC analysis as well as the ^1^H NMR and ^13^C NMR spectra showed that the VTMOS modified linseed oil was rapidly polymerized. After 3 h of linseed oil with VTMOS reaction, the polymer had a molecular weight similar to that of the unmodified linseed oil polymerized for 12 h. As can be easily observed, the silanisation of the linseed oil was much faster than in the case of the soybean oil silanisation reaction widely described in the literature. The silanisation of linseed oil did not require the use of cross-linking catalysts. The reason for such a high reactivity of linseed oil to VTMOS and the quick gelation of the resulting product is primarily the presence of numerous double bonds in linseed oil and their high availability. This applies in particular to the double bond in linolenic acid at the C16 carbon. Extending the reaction of linseed oil with VTMOS to more than 3 h led to further changes resulting from the hydrolysis and condensation reactions.

## Figures and Tables

**Figure 1 materials-13-05376-f001:**
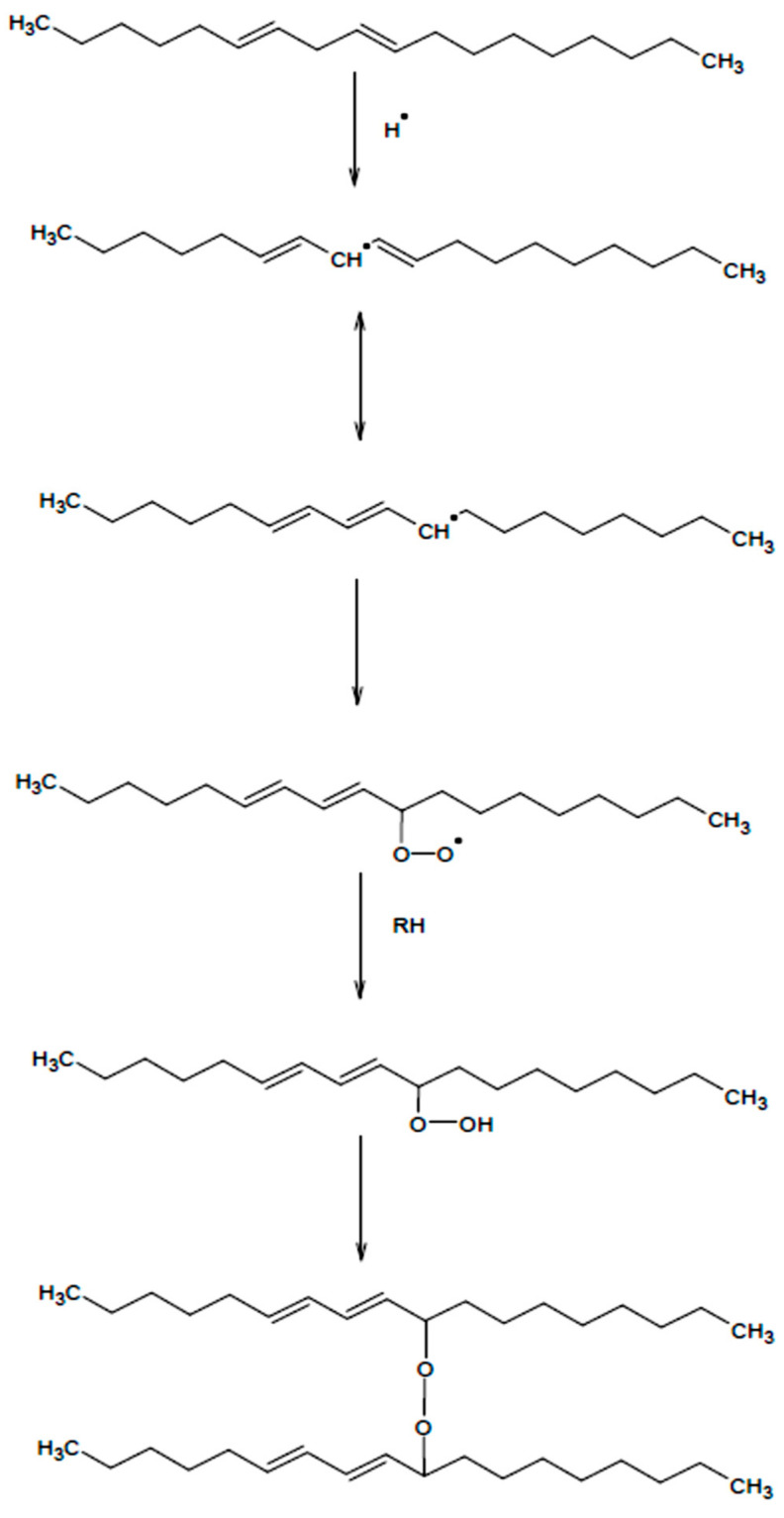
Mechanism of radical polymerisation of linolenic acid.

**Figure 2 materials-13-05376-f002:**
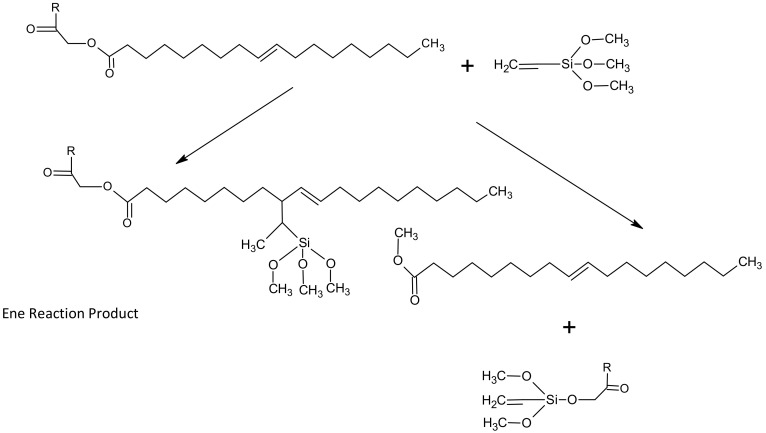
Alder “ene” reaction mechanism for soybean oil (diglycerides) and VTMOS.

**Figure 3 materials-13-05376-f003:**
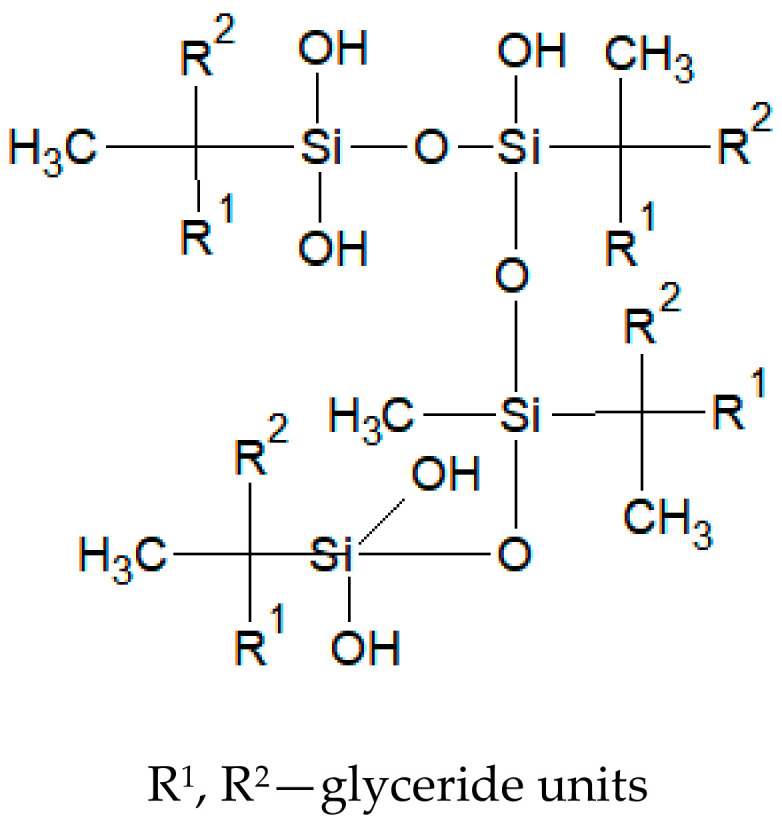
Reaction products of soybean oil and VTMOS after the introduction of water and dibutyltin dilaurate.

**Figure 4 materials-13-05376-f004:**
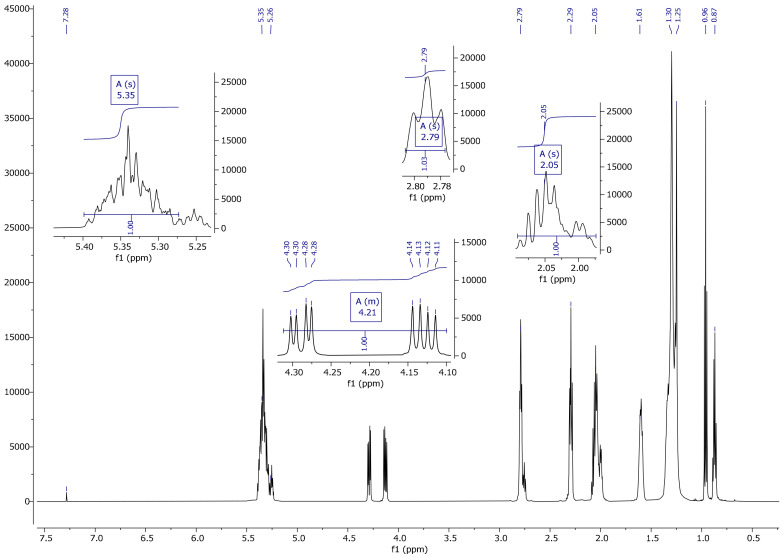
^1^HNMR spectrum of linseed oil.

**Figure 5 materials-13-05376-f005:**
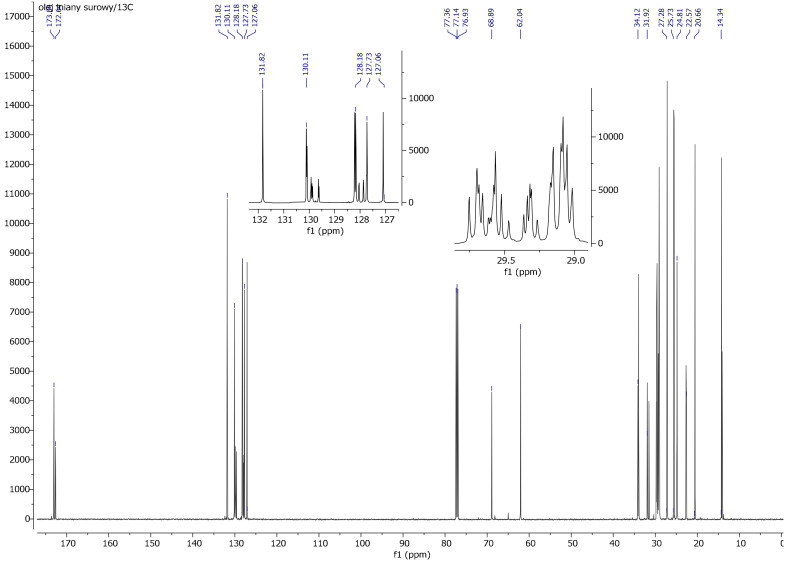
^13^C NMR spectrum of linseed oil.

**Figure 6 materials-13-05376-f006:**
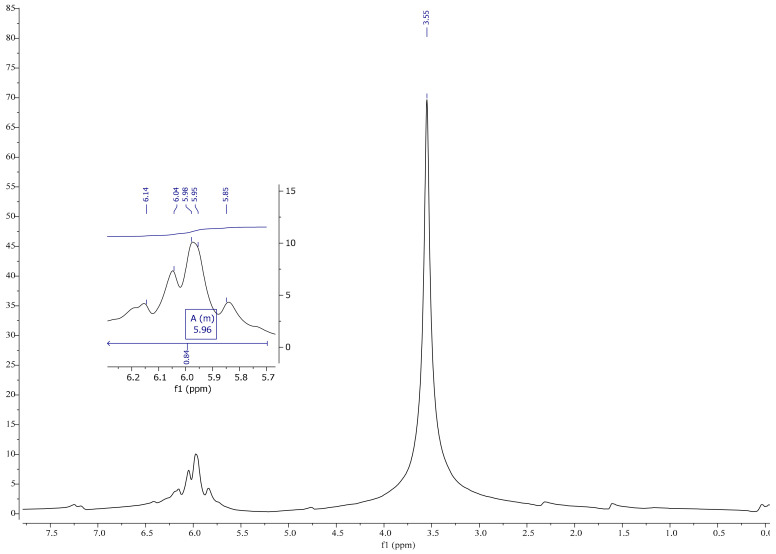
^1^H NMR spectrum of VTMOS.

**Figure 7 materials-13-05376-f007:**
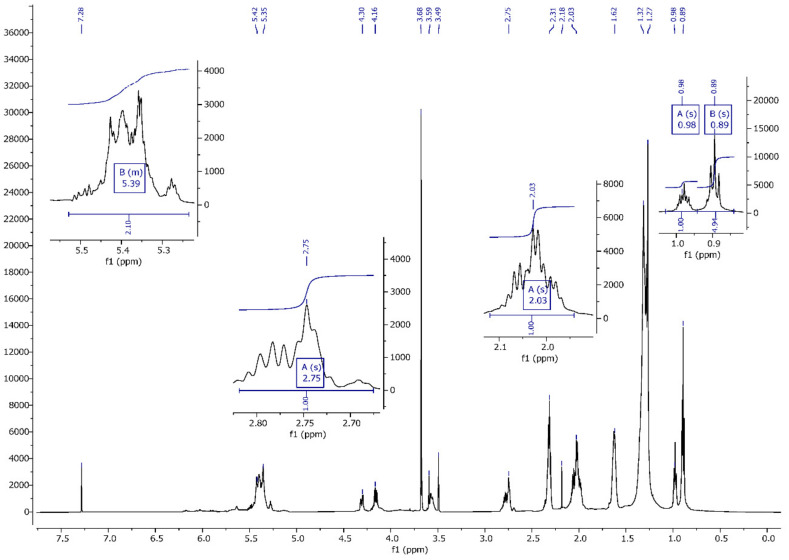
^1^H NMR spectrum of the linseed oil after reaction with VTMOS (reaction 4).

**Figure 8 materials-13-05376-f008:**
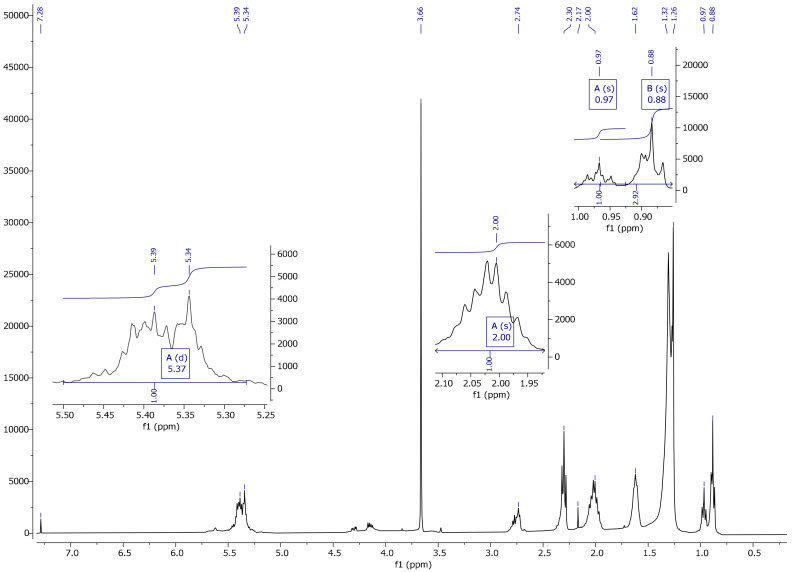
^1^H NMR spectrum of a gelled product from the reaction No. 3.

**Figure 9 materials-13-05376-f009:**
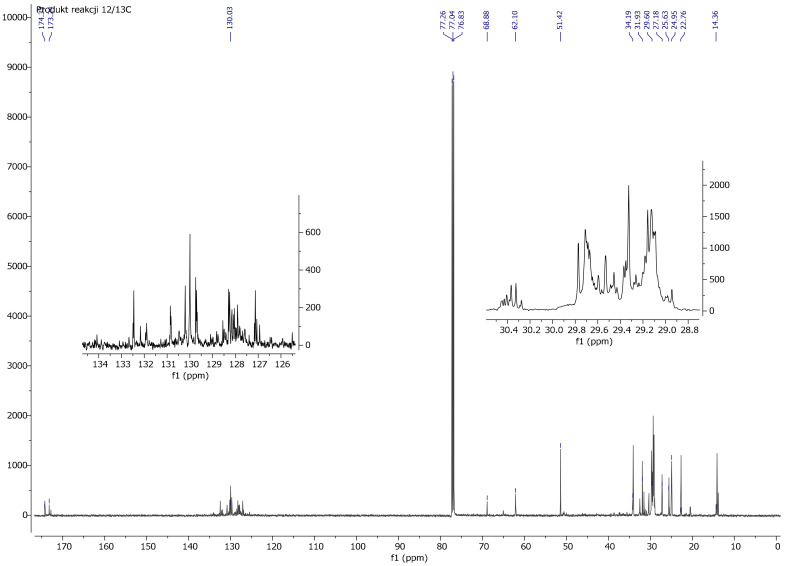
^13^C NMR spectrum of linseed oil after modification reaction using VTMOS (reaction No. 4).

**Figure 10 materials-13-05376-f010:**
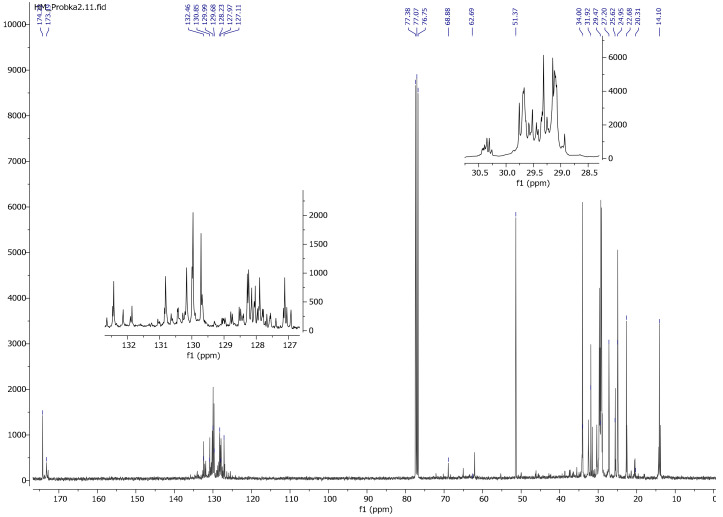
^13^C NMR spectrum of a gelled linseed oil after reaction with VTMOS (reaction No. 3).

**Table 1 materials-13-05376-t001:** The chemical composition of the reaction mixture based on linseed oil.

Reaction ID	Luperox 101 for 1 Mol of Oil [mol]	VTMOS for 1 Mol of Oil [mol]	Moment of VTMOS Addition	Reaction Time at 280 °C [h]
1	0	0	No addition	12
2	0.04	0	No addition	12
3	0.04	0.6	At the beginning of reaction	6
4	0.04	0.6	3rd h of reaction	6
5	0.04	1.2	At the beginning of reaction	3
6	0	1.2	At the beginning of reaction	12

**Table 3 materials-13-05376-t003:** Acid number for selected products.

Reaction ID	Acid Number [mg KOH/g]
1	4.59
2	6.14
3	0.47
4	0.23
5	0.47
Raw linseed oil	0.30

**Table 4 materials-13-05376-t004:** GC-FID analysis of substrates and products involved in silanisation reactions.

Sample	VTMOS Content [%]				Product Consistency
Reaction ID	0 h	3 h	6 h	12 h	
1	<0.02	<0.02	<0.02	<0.02	liquid
2	<0.02	<0.02	<0.02	<0.02	liquid
3	8	0.15	<0.02	–	gel
4	<0.02	8.1	0.3	–	liquid
5	15	3.2	–	–	liquid
6	15	2.8	0.09	<0.02	gel

**Table 5 materials-13-05376-t005:** GPC analysis for substrates and products involved in silanisation reactions.

Sample	0 h			3 h			6 h			12 h		
Reaction ID	Mn	Mw	PDI	Mn	Mw	PDI	Mn	Mw	PDI	Mn	Mw	PDI
1	1313	1413	1.08	1577	2203	1.45	1876	3320	1.77	2357	7036	3.90
2	1319	1561	1.18	1625	2653	1.63	1858	3315	1.79	2187	6064	2.77
3	1310	1677	1.28	1242	4804	3.86	1576	24459	15.52	–	–	–
4	1563	2744	1.75	1586	2579	1.62	1340	6186	4.61	–	–	–
5	1311	1611	1.23	1440	3879	2.69	–	–	–	–	–	–
6	1322	1421	1.07	1478	3711	3.62	1021	9269	9.08	714	2166	3.04

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
