# Peer review of "Influence of Reaction Parameters on the Gelation of Silanised Linseed Oil"

_materials, 2020, doi:10.3390/ma13235376_

Round 1

Reviewer 1 Report

The manuscript “Influence of reaction parameters on the gelation of silanised linseed oil” written by the authors Ewelina Depczyńska, Waldemar Perdoch and Bartłomiej Mazela describes the catalytic silylation reaction of the linseed oil with vinyltrimethoxysilane and an explanation of the reasons for rapid gelation of the reaction product was drawn. The subject of the work is suitable for publication in the journal, but some minor issues need to be solved before being accepted for publication. In the following points you will find enclosed a list of observations and suggestions that I consider indispensable for acceptance. After minor revision by the authors, the manuscript can be considered to be published in Materials.

In the abstract:

-rewrite “1H and 13C NMR” as “1H and 13C NMR”

- rewrite the phrase as it contains some mistakes: “The results confirm that after 3 hours of reaction of linseed oil with VTMOS in temp. 280℃, a molar mass of product reaction was similar to obtained as a result of the conventional polymerization.”

Lines 92-94 -  ” and an explanation of the reasons for rapid gelation of the reaction product.” please rewrite because the phrase appears incomplete.

Line 101 - “highpressure reactor” – correct to “high pressure reactor”

Line 103 – “the mixture stirring in room temperature” - correct to “the mixture stirring at room temperature”

In Table 1 – “Reaction time [h] in 280°C” – change to “Reaction time [h] at 280°C”

Line 136 – remove “gel permeation chromatography” and let only the abbreviation (GPC).

Lines 152 -153 – correct “13C spectra” and “13C NMR spectra” to 13C spectra and 13C NMR spectra, respectively

-The resolution of Figure 4, Figure 5, Figure 6 and Figure 7 needs to be improved for a better understanding.

Line 224 – “oil gelled” – change to “gelled oil”

Lines 238-240 – in the phrase ”In Linoleic acid, which is 53% soybean oil, the most reactive double bond is at the C13 carbon atom attached to the n-pentane  group (steric hindrance for radical polymerization)” , the linoleic acid is 53% in soybean oil or linseed oil? Because it is mentioned in a phrase above that linoleic acid is 7% in soybean oil. Please verify and correct if it is the case.

Line 257 - correct “1H NMR and 13C NMR spectra” to “1H NMR and 13C NMR spectra”

Please verify the references: 24, 29, 30, as they appear incomplete (number of pages and issue).

Reviewer 2 Report

This manuscript is written logically, and the background and significance are well discussed. It is an interesting manuscript. In this study, the authors demonstrated the influence of reaction parameters on the gelation of silanised linseed oil. It is an interesting manuscript which may has the guiding significance for the preparation of biomaterials.
However, there are still a few concerns that to be addressed.

Line 15 the abbreviations “VTMOS” should be added after "vinyltrimethoxysilane". 

Line 17 should give the full name of “NMR, GC-FID, GPC" when you use the abbreviation at the first place.

Line 19 "temp. " can be deleted because there already shows the unit oC.

Line 46 should give the full name of “UV/VIS" when you use the abbreviation at the first place.
